# Why Are Branched-Chain Amino Acids Increased in Starvation and Diabetes?

**DOI:** 10.3390/nu12103087

**Published:** 2020-10-11

**Authors:** Milan Holeček

**Affiliations:** Department of Physiology, Faculty of Medicine in Hradec Králové, Charles University, Šimkova 870, 50003 Hradec Králové, Czech Republic; holecek@lfhk.cuni.cz

**Keywords:** insulin, insulin resistance, pyruvate, glucose, alanine, obesity

## Abstract

Branched-chain amino acids (BCAAs; valine, leucine, and isoleucine) are increased in starvation and diabetes mellitus. However, the pathogenesis has not been explained. It has been shown that BCAA catabolism occurs mostly in muscles due to high activity of BCAA aminotransferase, which converts BCAA and α-ketoglutarate (α-KG) to branched-chain keto acids (BCKAs) and glutamate. The loss of α-KG from the citric cycle (cataplerosis) is attenuated by glutamate conversion to α-KG in alanine aminotransferase and aspartate aminotransferase reactions, in which glycolysis is the main source of amino group acceptors, pyruvate and oxaloacetate. Irreversible oxidation of BCKA by BCKA dehydrogenase is sensitive to BCKA supply, and ratios of NADH to NAD^+^ and acyl-CoA to CoA-SH. It is hypothesized that decreased glycolysis and increased fatty acid oxidation, characteristic features of starvation and diabetes, cause in muscles alterations resulting in increased BCAA levels. The main alterations include (i) impaired BCAA transamination due to decreased supply of amino groups acceptors (α-KG, pyruvate, and oxaloacetate) and (ii) inhibitory influence of NADH and acyl-CoAs produced in fatty acid oxidation on citric cycle and BCKA dehydrogenase. The studies supporting the hypothesis and pros and cons of elevated BCAA concentrations are discussed in the article.

## 1. Introduction

Branched-chain amino acids (BCAAs; valine, leucine, and isoleucine) are essential amino acids, which act as substrates and regulators of protein and energy metabolism and precursors to other amino acids. It has been repeatedly shown that their concentrations in blood plasma are uniquely increased during starvation and in both type 1 (T1DM) and type 2 (T2DM) diabetes mellitus [1,2,3,4,5]. However, although elevated BCAA levels in starvation and diabetes have been recognized for decades, and are receiving considerable attention, explanation of the pathogenesis of their increased levels is, surprisingly, not available.

In recent years, there has been an increasing number of studies on obese people demonstrating a strong relationship between BCAA levels and insulin resistance (IR), and considering increased BCAA levels relevant in predicting the development of T2DM [5,6,7]. Disturbances in BCAA metabolism have also been described in other IR states associated with future onset of T2DM, including cancer, burn injury, trauma, sepsis, kidney dysfunction, and fatty liver disease [7,8,9]. Much attention has been paid to the role of increased BCAA levels in the etiology of IR, intervention outcomes, and disease progression [10,11].

Since BCAA transamination, the first step in BCAA catabolism, occurs, unlike other amino acids, in skeletal muscle [12] and several studies have shown that increased BCAA levels in blood plasma in starvation and diabetes are associated with their increased concentrations in muscles [4,13,14,15,16], the muscles should play an important role in the pathogenesis of increased BCAA levels in plasma and some other tissues.

Both early starvation and T1DM are associated with depletion of insulin, increased proteolysis, and increased BCAA oxidation in muscles [16,17,18,19,20]. Hence, it may be supposed that increased BCAA levels are due to an insufficient increase in their catabolism to compensate for their increased influx from muscle proteins. In T2DM, where insulin resistance is a characteristic feature and there is no marked activation of proteolysis and BCAA oxidation, the cause of the increased BCAA levels should be the inhibition of their catabolism.

The aim of this article is to explore the hypothesis that impaired glycolysis and preferential use of fatty acids as a source of energy, characteristic features of starvation, and both types of diabetes [21,22,23], are the main causes of impaired catabolism of BCAAs in muscles resulting in increased BCAA levels in body fluids. First, I will provide a brief overview of BCAA catabolism, and an attempt will be made to describe the effects of glycolysis and fatty acid oxidation on BCAA catabolism in the muscles of healthy subjects. Second, I will explore the role of impaired glycolysis and increased fatty acid oxidation in pathogenesis of increased BCAA levels in starvation, T1DM, and T2DM.

## 2. Catabolism of BCAAs

Catabolism of BCAAs has several features in common. BCAA aminotransferase (BCAAT), the catalyst for reversible transamination of BCAAs to branched-chain keto acids (BCKAs), is the first enzyme in the catabolism of all three BCAAs. Each BCKA then undergoes irreversible decarboxylation by BCKA dehydrogenase (BCKAD) to the corresponding acyl-CoA esters. Then, catabolism diverges into separate pathways. Leucine is catabolized to acetyl-CoA and acetoacetate, valine to succinyl-CoA, and isoleucine to acetyl-CoA and succinyl-CoA. It is the consensus, that BCAAT and BCKAD play the main regulatory roles in BCAA catabolism.

### 2.1. Branched-Chain Amino Acid Aminotransferase (BCAAT)

There are two BCAAT isoenzymes, mitochondrial and cytosolic. The mitochondrial is expressed ubiquitously, the cytosolic is restricted to the brain, ovary, and placenta [24]. High activity of mitochondrial BCAAT is in skeletal muscle, and very low expression is in the liver [12]. Since the Km of BCAAT is two- to fourfold higher than tissue BCAA concentrations [25], the rate of transamination responds rapidly to changes in tissue BCAA availability. It has been suggested that insulin exerts its effect on BCAA catabolism primarily through regulation of protein metabolism and subsequent changes in BCAA availability [26].

The main acceptor of the amino nitrogen of BCAAs is α-ketoglutarate (α-KG), which is converted to glutamic acid (GLU). A portion of GLU produced in the BCAAT reaction is used for the synthesis of GLN in an irreversible reaction catalyzed by GLN synthetase. Conversion of GLU to α-KG by alanine aminotransferase (ALT) and aspartate aminotransferase (AST) reactions, in which pyruvate (PYR) and oxaloacetate (OA) are converted to alanine (ALA) and aspartate (ASP), attenuates the drain (cataplerosis) of α-KG from the citric acid (Krebs) cycle. Since BCAAT and the ALT and AST reactions are reversible and in equilibrium with their reactants [12,27,28], the competition among amino-donors (BCAA and GLU) and amino-acceptors (BCKA, α-KG, PYR, and OA) could influence the rate of transamination.

Most ALA and GLN synthesized in muscles are released into systemic circulation, whereas most of ASP are consumed in the purine nucleotide cycle and malate-aspartate shuttle. BCKAs are released into the blood by monocarboxylate transporters or oxidized by BCKAD (Figure 1).

### 2.2. Branched-Chain α-Keto Acid Dehydrogenase (BCKAD)

BCKAD is a mitochondrial multienzyme complex that catalyzes the irreversible decarboxylation of BCKA to branched-chain acyl-CoA esters. BCKAD activity is highest in the liver, intermediate in kidneys and heart, and low in muscles, adipose tissue, and brain [12]. Since BCKAD activity in muscles is much lower than BCAAT activity, most BCKAs produced in muscles are released into the bloodstream and are taken up by other tissues [12].

Long-term regulation of BCKAD activity is accomplished through changes in the expression of its subunits, short-term regulation of the complex occurs by reversible phosphorylation of its E1α subunit; a specific kinase inactivates, and a specific phosphatase activates. The main regulatory role is realized by BCKAD kinase, which is subject to inhibition by BCKA [29]. Therefore, increased flux of BCAAs through BCAAT due to increased BCAA supply after protein intake or breakdown of muscle proteins increases the complex activity and rate of BCAA oxidation [30,31]. In addition to phosphorylation, the flux of BCKA through BCKAD is inhibited by increased ratios of NADH to NAD+, acyl-CoA to CoA-SH, and ATP concentration [12].

## 3. Role of Glycolysis and Fatty Acid Oxidation in BCAA Catabolism

### 3.1. The Role of Glycolysis

Glycolysis, the metabolic pathway that converts glucose into PYR, is activated in the postprandial state (a state after meal intake, which lasts approximately 4 h) by increased glucose supply and insulin production. It has been shown that muscle tissue takes up 25–30% of an oral glucose load, from which ~50% is immediately oxidized, ~15% is released as potential gluconeogenic precursors, such as lactate, ALA, and PYR, and ~35% is stored [32]. Hence, glycolysis is the predominant fate of oral glucose taken up by muscles in the postprandial state.

Because BCAAT is almost absent in the liver, in the postprandial state, the BCAA concentrations increase in systemic circulation most of all amino acids [33,34]. The main locus for BCAA degradation is skeletal muscle, because of its mass and high BCAAT activity [12]. Several articles have demonstrated that food intake increases the flux through BCAAT resulting in increased BCAA oxidation and release of BCKA, GLN, and ALA from muscles [35,36,37,38].

It can be postulated that, in postprandial state, the activation of glycolysis is essential for the increased flux of BCAAs through BCAAT and the subsequent stimulatory influence of BCKA on BCKAD in three ways (Figure 2). First, glycolysis stimulates citric cycle activity [39,40] and subsequently increases the supply of α-KG for BCAAT. Second, in muscles, glycolysis is the exclusive source of PYR [41], which is used to attenuate the drain of α-KG from the citric cycle (cataplerosis) via the ALT reaction. Third, the citric cycle and PYR are sources of OA, which may also play a role in synthesis of α-KG from GLU via the AST reaction. It should be noted that OA acts as an essential substrate for irreversible reaction with acetyl-CoA to form citrate. In the presence of a low level of OA, the extent of metabolism via the Krebs cycle decreases [39,40].

The importance of glycolysis for BCAA catabolism is clearly evidenced by the most significant decrease in BCAA levels of all amino acids during the oral glucose tolerance test or euglycemic insulin clamp [42,43,44,45,46,47,48].

### 3.2. The Role of Fatty Acid Oxidation

Fatty acid oxidation (β-oxidation) is the mitochondrial process of breaking down a fatty acid into acetyl-CoA, an acyl-CoA derivative containing two carbons less than the original acyl-CoA molecule, and NADH. NADH production may decrease the activity of enzymes of the citric cycle that produce NADH (isocitrate dehydrogenase, α-KG dehydrogenase, and malate dehydrogenase). Increased ratios of NADH to NAD^+^ and of acyl-CoA to CoA-SH may decrease the flux of BCKA through BCKAD [12].

In postprandial state, fatty acid oxidation is suppressed; therefore, the inhibitory effects of NADH and acyl-CoA on the citric cycle and BCKAD activity are limited. However, several studies have shown that increased use of fat as a source of energy increases BCAA concentrations in the blood [10,49]. Fatty acid oxidation is activated in states characterized by decreased glycolysis, such starvation and in diabetes.

## 4. Etiopathogenesis of Increased BCAA Levels in Starvation, T1DM, and T2DM—Common Features

The main features of early starvation and diabetes are decreased glycolysis and increased oxidation of fatty acids due to insulin deficiency, decreased ratio of insulin to glucagon, and/or IR. It is hypothesized that just a decline in glycolysis and increased fatty acid oxidation in muscles cause alterations, which inhibit BCAA catabolism in muscles and subsequently increase BCAA concentrations in plasma and tissues (Figure 3). The main alterations include:*Impaired flux through the citric cycle due to decreased glycolysis and the inhibitory influence of NADH produced by fatty acid oxidation.* The consequence should be a decreased supply of α-KG for the BCAAT reaction.*Enhanced formation of NADH, acylcarnitines, and acyl-CoAs due to increased fatty acid oxidation.* The consequence should be decreased flux of BCKA through BCKA dehydrogenase. Increased BCKA levels have been reported in starvation and diabetes [13,50,51].*Decreased supply of OA and PYR for conversion of GLU to α-KG due to decreased glycolysis.* The consequence should be cataplerosis of α-KG and the shift of GLU metabolism towards GLN synthesis. The benefit for the body might be the use of GLN for ammonia synthesis by the kidneys and the subsequent increase in the buffering capacity of urine, which helps to compensate effects of increased production of ketone bodies on the acid-base balance during starvation and in T1DM. Increased GLN synthesis and expression of GLN synthetase in muscles have been reported in diabetic rats [52,53].*Impaired mitochondrial function.* Oxidative stress seen in diabetes and IR states increases the susceptibility of mitochondrial proteins to oxidative damage [54,55]. The consequences include decreased flux through the citric cycle and decreased activity of enzymes involved in BCAA catabolism.

Additional role in pathogenesis of increased BCAA levels might play:*Alterations in BCAA catabolism in the liver.* Since BCAAT is absent in the liver, increased breakdown of hepatic proteins due to starvation or diabetes may result in increased release of BCAAs from the liver to the blood [9,56]. This suggestion is supported by increased contents of BCAAs in the liver in both starvation and diabetes [13,15,56,57]. Alterations in hepatic BCKAD activity might also play a role. Both activation [58,59] and suppression [60,61,62] have been reported in rats with T1DM and T2DM. In my opinion, alterations in the liver are not sufficient to explain the pathogenesis of increased BCAA levels in starvation and diabetes. Just as occurs after food intake, enhanced amounts of the BCAA released from the liver to the blood should be efficiently removed by skeletal muscle if its metabolic functions are not impaired.*Alterations in BCAA catabolism in adipose tissue*. In adipose tissue, where leucine and isoleucine are substrates for fatty acid synthesis, insulin increases the activity of BCKAD [63,64]. Recent studies in obesity and IR have demonstrated downregulation of the expression of BCAA catabolizing enzymes in adipose tissue and suggest their role in the pathogenesis of increased BCAA levels [65,66,67,68]. However, considering that BCAAT activity in adipose tissue is much lower than in muscles and BCKAD activity is also very low [12,69], decreased BCAA oxidation in adipose tissue could have a minor role in the pathogenesis of increased BCAA levels in starvation and diabetes. The shortcomings of the hypothesis are also increased fat mass in obese people, which compensates for decreased activities of BCAA catabolic enzymes in adipocytes, and succinyl-CoA, the end-product of valine catabolism, which is not ketogenic. Therefore, adipose tissue may contribute to alterations in BCAA levels in insulin-deficient and IR states, but cannot play a major role.*Transamination of BCKAs to BCAAs.* Since the BCAAT reaction is reversible and near equilibrium, increased supply of BCKA and GLN from muscles, as occurs in starvation and diabetes, may shift the BCAAT reaction towards BCAA synthesis. An interorgan cycle, in which muscles act as a source of BCKA and most other tissues aminate the BCKA into the corresponding BCAA has been postulated [70,71,72].

## 5. Etiopathogenesis of Increased BCAA Levels in Starvation, T1DM, and T2DM—Specific Features

In this part of the article, I will report studies devoted specifically to starvation, T1DM, and T2DM to emphasize the differences in pathogenesis of alterations in glycolysis and fatty acid oxidation, differences in protein turnover, differences in origin of the BCAA intended for transamination and decarboxylation, and changes in BCKA, ALA, and GLN levels, which may indicate whether a decrease in glycolysis or activation of fatty acid oxidation plays a dominant role.

### 5.1. Why Are BCAAs Increased in Starvation?

Among the amino acids, starvation uniquely increases the concentrations of all three BCAAs. In humans, increases are evident within a day, and the maximum is reached within 3 days [1,2,45,50]. The first 3 days of starvation, called the early or adjustment phase of starvation, are characterized by a decreased ratio of insulin to glucagon, decreased glycolysis, increased gluconeogenesis in the liver, preferential use of fatty acids as a source of energy, and accelerated proteolysis in muscles. After 3 days of fasting, a phase called protein-sparing gradually develops, and the supply of BCAAs from protein degradation decreases. Increased BCAA levels persist for 8–10 days, and prolongation of starvation decreases BCAA below basal levels [1,2].

#### 5.1.1. Early Starvation

The only source of amino acids in starvation is the breakdown of endogenous proteins, especially in muscles. Whereas most proteinogenic amino acids released from muscle proteins during the early stage of starvation are efficiently oxidized or used for gluconeogenesis in the liver, most BCAAs must be catabolized in muscles. However, in early starvation, BCAA transamination and BCKA decarboxylation in muscles are inhibited due to alterations induced by decreased glycolysis and increased oxidation of fatty acids, as shown in the previous sections. Hence, BCAA catabolism in muscles is not sufficient to utilize the increased influx of BCAAs from increased protein breakdown. The result is increased BCAA levels in muscles, plasma, and tissues.

The postulation that the main role in the pathogenesis of increased BCAA levels in the early stage of starvation plays decreased glycolysis and increased oxidation of fatty acids in muscles is supported by:Higher increases in BCAA levels in muscles than in the plasma, liver, and heart of rats after 3 days of starvation [13].Increased BCAA and decreased GLN, GLU, and ALA concentrations in muscles of healthy volunteers after 72 h of fasting [73]. Results indicate impaired BCAA transamination.Increased release of BCAAs from forearm tissues to the blood in subjects fasted for 60 h [74].Decreased BCKAD activities in muscles of starving animals [75,76].Decreased ALA concentrations in blood plasma and muscles in starving subjects [77,78,79]. Results indicate impaired BCAA transamination in muscles.Incubation of muscles from 24 h-fasted chickens with acetoacetate and 3-hydroxybutyrate decreased glycolysis and PYR concentration, inhibited BCAA transamination and ALA formation, and increased GLU concentration and GLN release. The addition of PYR prevented the inhibitory effect of ketone bodies on BCAA transamination and ALA synthesis [78].Decreased ALA and GLN production by diaphragms obtained from 48 h-starved rats incubated with 3-hydroxybutyrate or acetate. Addition of PYR restored ALA and GLN production to control values [80].Parallel increments in BCAA and BCKA concentrations in blood plasma in the early stage of starvation [50]. This may reflect impaired BCAA transamination due to an insufficient rise in the flux of BCKA through BCKAD in muscles.

#### 5.1.2. Prolonged Starvation

The gradual decrease in plasma BCAA concentrations during prolonged starvation [1,2] reflects the effects of the protein-sparing period of starvation, in which energy expenditure decreases and nitrogen losses by urine are minimized. The BCAA levels are normalized and/or decreased in later stages of starvation due to gradual loss of muscle tissue, decreased turnover of muscle proteins, and subsequent decrease in BCAA appearance. Whereas BCKAD activity in muscles is decreased in the protein-sparing stage of starvation, marked increases in BCKAD activity occur in muscles and heart in the final stage of starvation, in which fatty acid oxidation decreases due to the loss of adipose tissue, and amino acids start to be the predominant energy substrate [76].

### 5.2. Why Are BCAAs Increased in T1DM?

The main characteristics of untreated T1DM include hyperglycemia, decreased glycolysis, activated gluconeogenesis, and dependency on fatty acids as a source of acetyl-CoA. In the presence of decreased supplies of OA and PYR, and excessive production of NADH by fatty acid oxidation, the extent of metabolism via the citric cycle and α-KG synthesis decrease [39]. Impaired entrance of acetyl-CoA into the citric cycle results in increased production of acylcarnitines, acyl-CoAs, and ketone bodies and acidosis development. Mitochondria isolated from the liver and muscles of subjects with T1DM showed decreased respiratory chain activity and ATP production, decreased expression of oxidative phosphorylation genes, increased ROS production, and decreased activity of citric cycle enzymes [54,55,81].

T1DM is also characterized by enhanced food intake, accelerated protein catabolism in muscles, and increased amino acid oxidation [16,59,82,83,84,85,86]. Hence, it is suggested that under conditions of decreased availability of amino group acceptors and impaired redox balance, flooding of the muscles by the BCAA originating from food and breakdown of muscle proteins greatly exceeds capacity of the body to catabolize the BCAA. The consequence is a marked increase in plasma BCAAs in patients with untreated T1DM [3,4,87]. In animals with T1DM, the BCAA concentration in blood plasma increases up to five-fold [16,88].

The hypothesis that increased BCAA levels in T1DM are due to disturbances in glycolysis and fatty acid oxidation resulting in an insufficient increase in BCAA catabolism in muscles is supported by:High BCAA levels in the muscles of patients with T1DM [4].High BCAA levels in the muscles of animals with experimental diabetes [13,14,15,16,89].Increased BCKA concentrations in blood plasma and muscles in rats with diabetes induced by alloxan [13]. Results indicate the role of impaired flux through BCKAD.Blunted activation of BCKAD by hyperleucinemia in muscles obtained from rats with streptozotocin-induced diabetes [16].Increased BCAA and decreased ALA concentration in blood plasma and muscles in animals with experimental T1DM [14,88]. Results indicate impaired BCAA transamination and conversion of GLU to α-KG.Decreased ALA concentrations in blood in patients with T1DM [3,11], which may be corrected by insulin therapy [11].Increased transamination and oxidation of BCAAs by isolated muscles of diabetic rats after the addition of PYR to medium [90,91]. Results suggest that flux through BCAAT is influenced by PYR availability.Decreased RNA amounts of BCAAT and BCKAD in the liver and muscles of rabbits with T1DM induced by alloxan [92].

### 5.3. Why Are BCAAs Increased in T2DM?

T2DM, characterized by hyperinsulinemia to compensate for insulin resistance, typically develops in association with obesity, hyperlipidemia, and prolonged physical inactivity. Relative increases in BCAA levels in the plasma of patients with T2DM are modest, e.g., ~10–25% [93] when compared with marked elevations of the BCAA in untreated T1DM [3,4,16,87,88].

Several studies have shown that plasma BCAA levels are also elevated in overweight and obese subjects, correlating positively with the degree of IR, and that the decline in BCAA levels is blunted in IR subjects during the oral glucose tolerance test [2,6,44,94,95,96]. It has even been suggested that increased plasma BCAA in obese subjects precedes alterations in glycemia and predicts the development of T2DM [5,95,97].

Unlike T1DM, protein turnover is unaltered in most patients with T2DM [86,98]; therefore, the supply of BCAAs to mitochondria for transamination is not increased or is increased only slightly. In addition, in contrast to T1DM, BCAA catabolism has been shown to be downregulated in T2DM [61,62]. Therefore, the pathogenesis of increased BCAA levels in obesity and T2DM can be easily explained by the inhibitory effects of impaired glycolysis and an excessive supply of fat on the transamination and decarboxylation of BCAAs in muscles. This hypothesis is supported by:Increased BCAA levels in muscles of animals with T2DM [51,57].Decreased BCKA and α-KG and increased acylcarnitine concentrations in muscles of humans with IR [99]. Decreased BCKA and α-KG indicate altered activity of BCAAT; increased acylcarnitine concentrations indicate incomplete oxidation of fatty acids.Increased BCKA concentrations in plasma, muscle, and the liver [51]. Results indicate impaired flux of BCKA through BCKAD.Decreased BCAAT activity in muscles, but not in liver and adipose tissue, of rats with IR induced by high fructose diet [69].Decreased expression of genes encoding BCAAT and BCKAD in the muscles of patients with T2DM [100].Lower acetate oxidation (measure of citric cycle activity) by myotubes isolated from T2DM subjects when compared with controls [101].

## 6. Consequences of Increased BCAA Concentrations

Several studies have shown that BCAA infusion worsens insulin sensitivity, and most papers published in recent years consider the increase of BCAAs in obese and diabetic people to be detrimental [5,6,7,10,11,102]. It is supposed that persistent activation of the mammalian target of rapamycin (mTOR) signaling pathway by increased BCAA concentrations plays a role in the pathogenesis of IR via interference with insulin signaling and increased degradation of insulin receptor substrates [10,103,104,105,106,107,108]. Articles published in recent years suggest that increased BCAA concentrations and related metabolites, such as medium- and long-chain acylcarnitines and various acyl-CoA species, are predictive of T2DM development, poor intervention outcomes, and disease progression [10,11,109,110,111,112].

Unfortunately, little attention is currently being paid to the potentially positive effects of increased concentrations of BCAAs, particularly in terms of protein balance, although for this reason BCAAs are recommended as supplements in muscle wasting disorders [9,113]. Moreover, all three BCAAs, especially leucine, stimulate insulin secretion and, in this way, may lower glycemia [114]. Leucine administration has been shown to improve protein balance in diabetic rats and adolescents with T1DM [115,116]. Finally, increased BCAA concentrations might play a role in attenuated breakdown of muscle proteins during the protein-sparing stage of starvation. Not quite clear are the consequences of increased plasma BCAA concentrations on brain uptake of other large neutral amino acids, particularly phenylalanine, tyrosine, methionine, histidine, and tryptophan, which share the same to transporter with the BCAAs.

## 7. Conclusions

Combined data from animal and human studies show that the main roles in the pathogenesis of increased BCAA levels in starvation and diabetes have decreased glycolysis and increased fatty acid oxidation in muscles. The main alterations include impaired BCAA transamination due to decreased availability of amino group acceptors, specifically α-KG, PYR, and OA, and impaired flux of BCKA through BCKAD due to excess NADH and an increased ratio of acyl-CoA to CoA-SH. Less important in the pathogenesis of increased BCAA levels is their reduced degradation in liver and adipose tissue. Systematic studies are necessary to assess the pros and cons of elevated BCAA levels in insulin-deficient and IR states.

## Figures and Tables

**Figure 1 nutrients-12-03087-f001:**
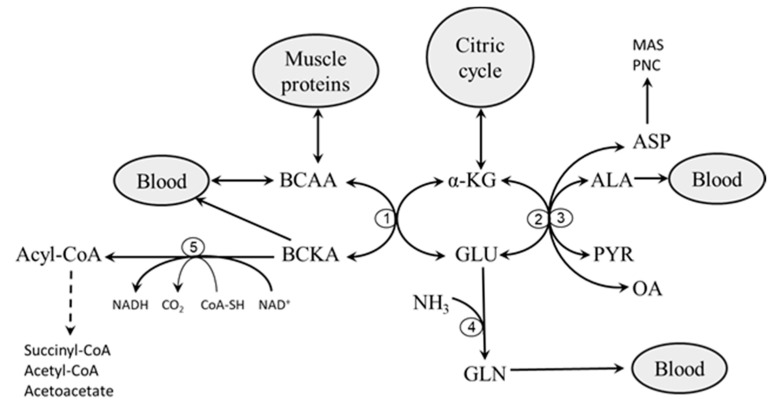
Main pathways of branched-chain amino acid (BCAA) catabolism in muscles. 1, BCAAT; 2, ALT; 3, AST; 4, glutamine synthetase; 5, BCKAD. ALA, alanine; ASP, aspartate; BCKA, branched-chain keto acids; CoA-SH, coenzyme A; GLU, glutamate; GLN, glutamine; OA, oxaloacetate; PYR, pyruvate; MAS, malate-aspartate shuttle; PNC, purine nucleotide cycle. α-KG, α-ketoglutarate.

**Figure 2 nutrients-12-03087-f002:**
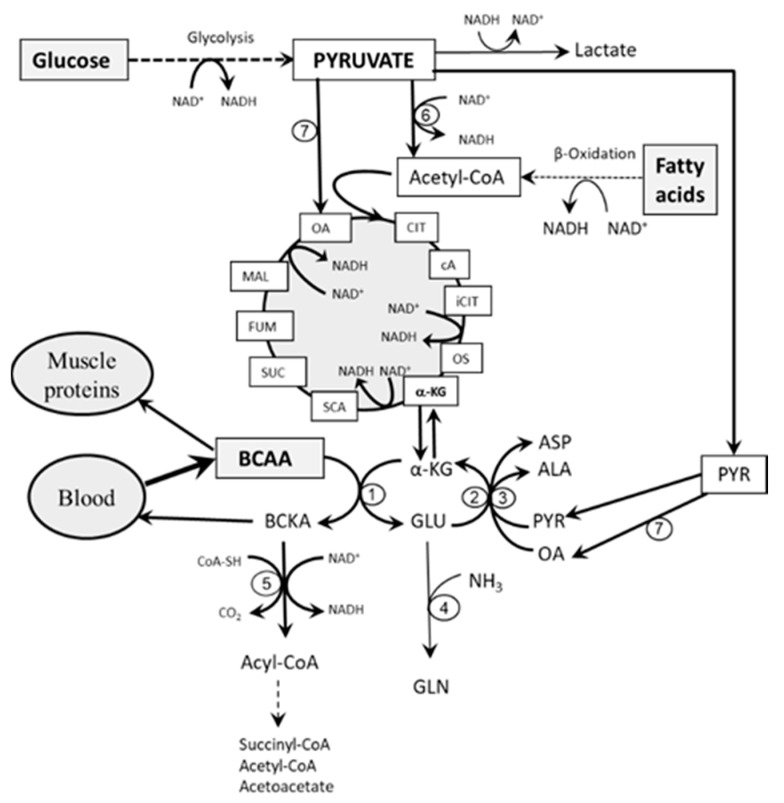
Effects of glycolysis on BCAA catabolism in muscles. Glycolysis ensures the supply of α-ketoglutarate (α-KG) for branched-chain amino acid aminotransferase (BCAAT) via a stimulatory influence on the citric cycle and supplying PYR and OA for alanine aminotransferase (ALT) and aspartate aminotransferase (AST) reactions. Branched-chain keto acids (BCKA) produced by BCAAT activate their own flux through BCKAD and/or are released to the blood. 1, BCAAT; 2, ALT; 3, AST; 4, glutamine synthetase; 5, BCKAD; 6, pyruvate dehydrogenase; 7, pyruvate carboxylase. ALA, alanine; ASP, aspartate; cA, cis-aconitate; CIT, citrate; CoA-SH, coenzyme A; iCIT, isocitrate; FUM; fumarate; GLU, glutamate; GLN, glutamine; MAL, malate; OA, oxaloacetate; OS, oxalosuccinate; PYR, pyruvate; SCA, succinyl-CoA; SUC, succinate. α-KG, α-ketoglutarate.

**Figure 3 nutrients-12-03087-f003:**
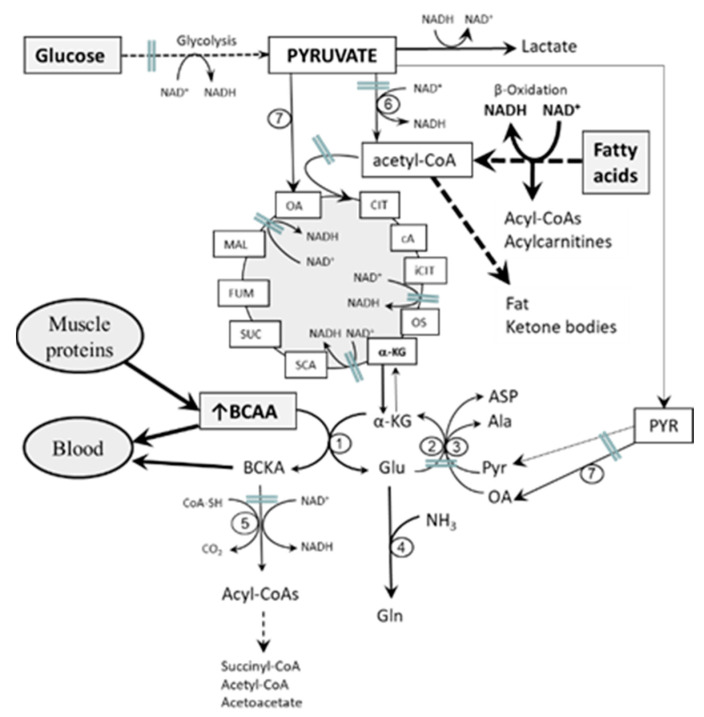
Role of skeletal muscles in the etiopathogenesis of increased BCAA levels in starvation and diabetes. Starvation and diabetes are characterized by decreased glycolysis and the preferential use of fatty acids as an energy source. The consequences are (i) decreased flux through the citric cycle, (ii) decreased supply of amino group acceptors (α-KG, OA, and PYR) for BCAAT and ALT and AST reactions, (iii) excessive production of NADH and acyl-CoAs with different lengths of carbon chain due to activated, but incomplete, oxidation of fatty acids, and, subsequently, (iv) decreased flux through BCKAD. 1, BCAAT; 2, ALT; 3, AST; 4, glutamine synthetase; 5, BCKAD; 6, pyruvate dehydrogenase; 7, pyruvate carboxylase.

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
