# Peer review of "Why Are Branched-Chain Amino Acids Increased in Starvation and Diabetes?"

_nutrients, 2020, doi:10.3390/nu12103087_

Round 1
Reviewer 1 Report
The current review was designed to address a question of clinical interest, the increase of BCAA in starvation and diabetes. This is a well organised, well illustrated, clear and well written manuscript which makes an important contribution to a very interesting topic, regarding biochemical pathways contributing to increased BCAA levels regarding diabetes and starvation from published data among human and animal studies.
Author Response
Thank you very much for the positive evaluation of my article.
Milan Holeček
Reviewer 2 Report
The author has analyzed the information available regarding the etiopathogenesis of increased BCAA in diabetes and starvation. Several different angles linked to the increase of BCAA level in the plasma, muscle, liver and other tissues have been looked at and a few speculations or hypotheses have been proposed, which are quite helpful audiences to understand the causes and impacts of increased BCAA levels, and the alterations at different time or stages of starvations and postprandial times. The author has explained step by step in a clear logic flow on the mechanisms and conditions that result in or associated with the increase of BCAA levels. The manuscript is well written and has offered valuable information and opinions through comprehensive analysis of literatures. It appears that the majority of references cited are those published before 2010, and many papers including reviews published more recently have not been included. The author may improve the manuscript by including the reported new findings and opinions available from the most recently published papers.
Author Response
Thank you very much for the positive evaluation of my work. Citations of recent articles, which may emphasize importance of increased concentration of the BCAA for clinical practice, have been included into the revised version of my article. The following references have been added:
- Lynch, C.J.; Adams, S.H. Branched-chain amino acids in metabolic signalling and insulin resistance. Nat. Rev. Endocrinol. 2014, 10, 723-736.
- Siddik, M.A.B.; Shin, A.C. Recent progress on branched-chain amino acids in obesity, diabetes, and beyond. Endocrinol. Metab. (Seoul) 2019, 34, 234-246.
- Lu, J.; Xie, G.; Jia, W.; Jia, W. Insulin resistance and the metabolism of branched-chain amino acids. Front. Med. 2013, 7, 53-59.
- Giesbertz, P.; Daniel, H. Branched-chain amino acids as biomarkers in diabetes. Curr. Opin. Clin. Nutr. Metab. Care 2016, 19, 48-54.
- Yoon, M.S. The emerging role of branched-chain amino acids in insulin resistance and metabolism. Nutrients 2016, 8, 405.
- Arany, Z.; Neinast, M. Branched Chain Amino Acids in Metabolic Disease. Curr. Diab. Rep. 2018, 18, 76.
Milan Holecek